Bulk and single-cell RNA sequencing identify prognostic signatures related to FGFBP2+ NK cell in hepatocellular carcinoma

Wu Yinbing 1
Peng Huanjun 1
Chen Guangkang 1
Tu Yinuo tuyinuo123@126.com 1
Yu Xinpei yxinpei@gzhmu.edu.cn 2
1 Department of Hepatobiliary Surgery, Affiliated Cancer Hospital & Institute of Guangzhou Medical University , Guangzhou , China
2 Department of Oncology, Affiliated Cancer Hospital & Institute of Guangzhou Medical University , Guangzhou , China
Guan Fanglin
Electronic publication date: 2025 May 20
Publication date: 2025
Volume: 13
Electronic Location ID: e19337
Received 2025 Jan 7; Accepted 2025 Mar 27
Copyright: ©2025 Wu et al.
Copyright year: 2025
Copyright holder: Wu et al.
License: This is an open access article distributed under the terms of the Creative Commons Attribution License, which permits unrestricted use, distribution, reproduction and adaptation in any medium and for any purpose provided that it is properly attributed. For attribution, the original author(s), title, publication source (PeerJ) and either DOI or URL of the article must be cited.
License URL: https://creativecommons.org/licenses/by/4.0/

Keywords: Hepatocellular carcinoma, FGFBP2+ NK cell, Prognostic signature, Tumor microenvironment, Immunotherapy, Bulk RNA sequencing, Single-cell RNA sequencing

Funding: Science popularization special project of Department of Science and Technology of Guangdong Province 2023A1414020055 This study was supported by Science popularization special project of Department of Science and Technology of Guangdong Province (2023A1414020055). The funders had no role in study design, data collection and analysis, decision to publish, or preparation of the manuscript.

==============================
Background

Hepatocellular carcinoma (HCC) is a highly aggressive malignancy. As a specific immune cell subpopulation, FGFBP2+ NK cells play a crucial part in immune surveillance of HCC progression. This study set out to identify prognostic signature related to FGFBP2+ NK cell in HCC.

Methods

Bulk and scRNA-seq data were derived from the public databases. The single cell atlas of HCC and heterogeneity of natural killer (NK) cells were delineated by “Seurat” package. Pseudo-time trajectory of FGFBP2+ NK cell was constructed by “Monocle2” package. Cell-cell interactions were analyzed by “CellChat” package. Prognostic signature was screened to develop a RiskScore model, and the prediction robustness was verified. Immune cell infiltration and immunotherapy response were assessed between different risk groups. Drug sensitivity was predicted by “oncoPredict” package. The expressions of the prognosis gene signature were detected by in vitro test utilizing HCC cells. The effects of key genes on the proliferative, migratory and invasive capacity of HCC cells were assessed by EdU assay, wound healing and Transwell assay.

Results

The proportion of NK cell in HCC samples was markedly decreased than that in healthy samples. NK cell was further divided into three cell subpopulations, and FGFBP2+ NK cell was associated with the prognosis of HCC patients. Pseudo-time trajectory analysis of FGFBP2+ NK cell revealed two differential expression gene clusters. FGFBP2+ NK cell exhibited extensive intercellular communication in HCC. Further, eight prognostic signatures were identified, including six “risk” genes (UBE2F, AHSA1, PTP4A2, CDKN2D, FTL, RGS2) and two “protective” genes (KLF2, GZMH). RiskScore model was established with good prognostic prediction performance. In comparison to low-risk group, high-risk group had poorer prognosis, lower immune cell infiltration, and higher TIDE score. Moreover, 16 drugs showed significant correlation with RiskScore. Additionally, the expressions of GZMH was downregulated while FTL, PTP4A2, UBE2F, CDKN2D, RGS2, and AHSA1 were up-regulated in HCC cells. FTL and PTP4A2 silencing could suppress the proliferation, migration and invasion abilities of HCC cells.

Conclusion

This study identified eight prognostic gene signatures related to FGFBP2+ NK cell in HCC, which may serve as potential therapeutic targets for HCC.

Introduction

Hepatocellular carcinoma (HCC) accounts for 80–90% of all primary liver cancer cases (Li, Liu & Qin, 2022; Zhou et al., 2024) and is the fifth most frequently diagnosed malignancy and third highest in carcinoma-relevant mortality around the world (Zhao et al., 2019; Li et al., 2024d; Sensi et al., 2024). It is generally believed that the critical risk factors leading to HCC primarily comprise hepatitis B/C virus infection, non-alcoholic fatty liver, and excessive alcohol consumption (Wang et al., 2021; Cao, Hu & Tang, 2023). Owing to the lack of obvious clinical symptoms and effective intervention strategies, the treatment outcomes for HCC have constantly attracted much attention (Kong & Yao, 2021). For the moment, the detection of HCC in clinical is widely dependent on the imaging approaches such as ultrasonography, computerized tomography, and magnetic resonance (Zhou et al., 2020b), as well as tumor biomarkers (particularly α-fetoprotein) (Wu et al., 2022). However, the specificity and sensitivity of these screening techniques are far from satisfactory. As a consequence, most patients with HCC have progressed into advanced stage when diagnosed (Du et al., 2021). In addition, despite chemotherapy, radiotherapy, targeted therapy, and immunotherapy have recently made remarkable progression (Liu et al., 2023), the prognostic outcomes of HCC remains disappointing due to the high recurrence and metastasis rate, with the 5-year survival probability <30% (Zhu et al., 2023b). Therefore, it is essential to develop novel prognostic signatures and offer promising therapeutic targets for HCC.

Researches have manifested that the prognosis of HCC patients is usually affected by the complicated tumor microenvironment (TME) (Zhu et al., 2023a). In recent years, immunotherapy such as immune checkpoint inhibitors (ICIs) have shown great promise in the treatment of HCC (Huang et al., 2021a). Whereas, one of the main barriers to effective immunotherapy is TME, which may contribute to immune tolerance and evasion in HCC (Luo et al., 2022). Comprehensive excavation of the concealed information in TME is critical for understanding the pathogenesis of HCC and improving the patient prognosis (Wu et al., 2024). Natural killer (NK) cell belongs to a member of innate immunity and is primarily found in human liver (Shin et al., 2024). NK cell has strong cytotoxicity and immunosurveillance potential, exerting crucial roles in the first-line immune defense against HCC development (Hong et al., 2022). The dysfunction of NK cell is considered to be a vital mechanism of immune escape in HCC (Sung & Jang, 2018). It has been confirmed that compared with healthy individuals, the proportion of NK cell is markedly decreased in the patients with advanced HCC (Hosseinzadeh et al., 2018). Immune cell therapy targeting NK cell has been emphatically recognized as the novel standard of care for advanced HCC (Hosseinzadeh et al., 2019). Given the important immune regulation roles of NK cell in HCC, in-depth exploration of NK cell at molecular levels is likely to facilitate the discovery of new immunotherapy strategies.

The single-cell RNA sequencing (scRNA-seq) technique can reveal the heterogeneity and dynamic changes of specific cells in TME, while bulk RNA sequencing can provide the whole transcriptomic information of tumor samples (Chen et al., 2024; Chen, Lin & Luo, 2024). In this study, by integrating bulk and scRNA-seq analysis, we revealed the single cell atlas of HCC and the heterogeneity of NK cell. Further, we analyzed the correlation between NK cell (especially FGFBP2+ NK cell) with the prognostic outcomes of HCC. A prognostic gene signature was then identified to construct RiskScore model, and the immune cell infiltration, as well as immunotherapy response was assessed between different risk groups. In addition, the impacts of prognostic signatures on HCC cell proliferation, migration and invasion were assessed by in vitro validation tests. We hope that this study could provide novel targets for immunotherapy in HCC, thereby enhancing treatment effectiveness and improving survival rates.

Material and methods

Data collection and preprocessing

RNA sequencing data and clinical follow-up data of HCC samples were downloaded from The Cancer Genome Atlas (TCGA) database through Genomic Data Commons (GDC) Application Programming Interface (API). The ICGC-LIRI-JP dataset was collected from the HCCDB database. Then, after deleting samples that did not have clinical follow-up data or status, the Ensembl was transformed to a gene symbol, and the expression average value was taken for multiple gene symbols. Finally, 370 tumor samples and 50 control samples were acquired in TCGA-LIHC cohort, utilizing as the training set. The ICGC-LIRI-JP cohort contained 212 HCC samples, which was served as the validation set.

The scRNA-seq data of GSE162616 dataset, including three HCC samples and three healthy liver samples, was derived from the Gene Expression Omnibus (GEO) database. For filtering the scRNA-seq data, each gene was set to be expressed in a minimum of three cells, and each cell expressed at least 200 genes. Then, the cells with nFeature_RNA >300, nCount_RNA >3,000, and mitochondrial gene expression percent.mito <25% were reserved. The NormalizeData function was applied for log conversion, and the FindVariableFeatures function was used to identify highly variable genes. Next, the ScaleData function was utilized to normalize the expression values of all genes, the RunPCA function was employed for principal component analysis (PCA), and the “harmony” R package was used to remove batch effects between different samples (Zulibiya et al., 2023).

Cell clustering analysis

Cell clustering analysis of GSE162616 dataset was conducted using the “Seurat” R package (Du et al., 2024). Firstly, UMAP was conducted on the top 15 principal components (PCs) for dimensionality reduction. Then, the cells were clustered using the FindNeighbors and FindClusters functions, with cluster resolution of 0.1 for all cells and 0.3 for NK cells. Finally, cell types were annotated based on the marker genes provided by CellMarker2.0 database (Lu et al., 2024).

Single-sample gene set enrichment analysis

The correlation between the prognosis of HCC and each type of NK cells was examined based on the single-sample gene set enrichment analysis (ssGSEA) score calculated utilizing “GSVA” R package for the tumor samples in TCGA-LIHC cohort (Wang et al., 2023). The surv_cutpoint function was applied to find the optimal cut-off point, and HCC patients were divided into low- and high-score groups.

Pseudo-time trajectory analysis

In order to invesitgate the role of FGFBP2+ NK cell in the progression of HCC, Monocle2 was used to construct pseudo-time trajectory (Tao et al., 2024). The cds object was established by newCellDataSet function, and the genes expressed in less than 10 cells were filtered out. The differentially expressed genes (DEGs) between HCC samples and healthy samples were identified by differentialGeneTest function, and Kyoto Encyclopedia of Genes and Genomes (KEGG) enrichment analysis was conducted on these DEGs using the “ClusterProlifer” R package (Zhang et al., 2024). Subsequently, the reduceDimension function was employed for dimensionality reduction (max_components = 2, method = “DDRTree”), and the orderCells function was applied to order the cells and construct the pseudo-time trajectory of FGFBP2+ NK cell.

Cellular communication analysis

The “CellChat” R package was employed to conduct cell–cell interaction analysis in HCC (Lian et al., 2024). The number of interactions and interaction weights/strength was analyzed, as well as the key ligand–receptor pairs between FGFBP2+ NK cell and other cell subpopulations were visualized by a bubble plot.

Establishment and verification of RiskScore model

Univariate Cox regression analysis (p < 0.05) was performed on the marker genes of FGFBP2+ NK cell. The number of genes in the model was reduced by LASSO Cox regression analysis was conducted using the “glmnet” R package (Zeng & Chen, 2024). By 10-fold cross validation, we selected the optimal lambda value as the result of LASSO regression for subsequent analysis. Further, stepwise regression analysis was performed, and the prognostic gene signature related to FGFBP2+ NK cell in HCC was identified to establish RiskScore model. The RiskScore of each patient in TCGA-LIHC cohort was obtained according to the following formula (Li et al., 2024a): RiskScore= ∑βi*ExPi

β is the coefficient of gene in Cox regression model, and i is the gene expression.

Z-score was utilized for standardization, and based on the threshold of RiskScore = 0, high- and low-risk groups were classified. To evaluate the prognostic prediction performance of RiskScore model, receiver operating characteristic (ROC) analysis was conducted using the “timeROC” R package (Lin et al., 2023). Additionally, the robustness of RiskScore model was validated in the ICGC-LIRI-JP cohort.

Immune cell infiltration and immunotherapy response analysis

Immune cell infiltration was assessed between high- and low-risk groups in TCGA-LIHC cohort. StromalScore, ImmuneScore, and ESTIMATEScore were calculated by the “estimate” R package (Dong et al., 2023). Tumor Immune Estimation Resource (TIMER) was applied to assess the infiltration of six immune cells (Xiao et al., 2020). Microenvironment cell populations-counter (MCP-counter) algorithm was utilized to assess the infiltration of 10 immune cells (Chen et al., 2022).

The response of different risk groups to immunotherapy was predicted by TIDE algorithm (Li et al., 2024b). Exclusion, Dysfunction, and TIDE scores of different risk groups were calculated in TCGA-LIHC cohort. Moreover, we analyzed the relationship of RiskScore and 9 immune checkpoint genes.

Correlation analysis between RiskScore and drug sensitivity

The IC50 values of drugs for HCC patients in TCGA-LIHC cohort were predicted using the “oncoPredict” R package (Maeser, Gruener & Huang, 2021). Then, the association between RiskScore and drug sensitivity was analyzed (p < 0.05 and |cor| > 0.3).

Cell culture and transfection

Human hepatic astrocyte cell line LX-2 (BNCC337957) and HCC cell line Huh7 (BNCC337690) were acquired from the BeNa Culture Collection (BNCC) Biotechnology Co. (Beijing, China). Then, LX-2 and Huh7 cell lines were separately cultivated in RPMI-1640 (BNCC338360) and DMEM (BNCC363314) contained 10% fetal bovine serum (FBS), and were all incubated in the condition of 5% CO2 and 37 °C. Hereafter, according to the protocol of Lipofectamine 2000 (Invitrogen, Waltham, MA, USA), Huh7 cells were transfected with the small interfering (si) RNA of FTL (si-FTL: 5′-TCCCAGATTCGTCAGAATTATTC-3′, Sangon, China), (si) RNA of PTP4A2 (si-PTP4A2: 5′-GAGGTTCTATGTGCCATAATTAA-3′, Sangon, China) and negative control (si-NC). We have performed short tandem repeat (STR) identification on the cells, and the mycoplasma detection results turned out to be negative.

Quantitative real-time PCR

Total RNA of LX-2 and Huh7 cells were acquired employing the Trizol reagent (B610409, Sangon, China). Then, the complementary DNA (cDNA) was synthesized through reverse transcription applying the RevertAid First Strand cDNA Synthesis Kit (B300538, Sangon, China). Thereafter, quantitative real-time PCR (qRT-PCR) amplification was performed using the SYBR Green (B110031, Sangon, China) on the basis of manufacturer’s instructions. The qPCR conditions were: 94 °C for 30 s first, then 40 cycles of 94 °C for 5 s and 60 °C for 30 s. The primer pairs of this study were shown in Table S1. GAPDH was utilized as an internal control to normalize the relative mRNA expressions of each gene by 2−ΔΔCT method (Xu et al., 2020b).

Proliferation assay

Huh-7 cells that had been transfected were maintained for 48 h until they reached the logarithmic growth phase, after which they were moved into 96-well plates. The EdU Cell Proliferation Assay Kit (RiboBio, Guangzhou, China) was utilized to evaluate cell proliferation. In accordance with the established protocol, the cells underwent staining, followed by examination and imaging using a fluorescence microscope (Nikon, Toyko, Japan). EdU-positive cell counts were conducted using ImageJ software.

Wound healing assay

The effect of FTL and PTP4A2 silencing on the migration of HCC cells Huh7 was measured by wound healing assay (Song et al., 2021). The transfected Huh7 cells (1 × 105) were inoculated into a 6-well plate and grown overnight. Afterwards, the wound was created using a sterile 20 µL pipette tip and Huh7 cells were sustainably cultured in serum-free medium for 48 h. Finally, the pictures of wound areas at 0 hour (h) and 48 h were obtained under a microscope (ECLIPSE Ei, Nikon, Tokyo, Japan), and the wound closure (%) of Huh7 cells was estimated with the ImageJ2 software.

Transwell assay

Transwell assay was conducted to examine the influence of FTL and PTP4A2 silencing on the invasion of HCC cells Huh7 (Huang et al., 2021b). The diluted Matrigel (BD Biosciences, Franklin Lakes, NJ, USA) was pre-coated into the Transwell chamber (8.0 µm, Corning, Corning, NY, USA). Next, the transfected Huh7 cells (1 × 105) were starved in the upper chamber encompassing 200 µL non-serum medium. 500 µL DMEM containing 10% FBS was filled into the lower chamber. After 48 h of incubation, the invaded Huh7 cells were fixed by 4% paraformaldehyde (YTB1299, bjbalb, Beijing, China) and dyed by 0.1% crystal violet (YT913, bjbalb, Beijing, China). Further, the number of invaded Huh7 cells was counted under the same microscope as above.

Statistical analysis

The bioinformatic analysis was conducted using R programing language (version 4.1.0). The differences between different groups were compared by the Wilcoxon rank-sum test. Kaplan–Meier (K-M) analysis of overall survival (OS) was conducted by log-rank test. The Spearman method was utilized for correlation analysis. All experimental data of independent triplicates were expressed as mean ± standard deviation,, and statistical analysis was carried out by GraphPad Prism8.0. For data that did not conform to a normal distribution, non-parametric tests such as the Mann–Whitney U test or Kruskal-Wallis test were applied. p < 0.05 signified statistical significance.

Results

Single-cell atlas of HCC revealed the reduction proportion of NK cell

The scRNA-seq data of GSE162616 dataset was analyzed to delineate the single-cell atlas of HCC. After cell filtration, standardization, dimensionality reduction, and clustering, a total of 47,550 cells were obtained and divided into 11 cell clusters (Fig. S1). Using the marker genes, these cells were further annotated as seven cell types (Fig. 1A), comprising NK cell (GZMB, GZMH, FGFBP2, GNLY, CD160), T cell (CD2, CD3D, CD3E, IL7R), macrophage (AIF1, MS4A7, LYZ), plasma cell (DERL3, IGHG1, MZB1), CD8+ T cell (MKI67, STMN1), B cell (CD79A, MS4A1), and hepatocyte (ALB, APOA1, KRT18) (Fig. 1B). Additionally, the proportion of each cell type in different samples was analyzed, showing that the number of NK cell in HCC samples was markedly decreased compared with healthy samples (Figs. 1C and 1D). These results indicated that the proliferation and activity of NK cell may be inhibited in HCC microenvironment.

Figure 1 Single-cell atlas of hepatocellular carcinoma (HCC).

(A) UMAP plot of cell subpopulations after annotation; (B) Expression level of marker genes in each cell type; (C–D) Proportion of each cell type in different samples.

FGFBP2+ NK cell was associated with the prognosis of HCC patients

NK cell as an important immune cell can recognize and kill tumor cells. Hence, we extracted the NK cell from HCC tissues for re-clustering, obtaining three cell subpopulations (Fig. 2A). The top20 highly expressed genes in each NK cell subpopulation were displayed by a heatmap (Fig. 2B). Using the marker genes, these cell subpopulations were annotated as FGFBP2+ NK cell (FGFBP2, FCGR3A, GZMB, GZMH), CD160+ NK cell (GZMK, CD160), and IL7R+ NK cell (IL7R, SELL, LMNA, CD44) (Fig. 2C). Compared with healthy samples, the proportion of FGFBP2+ NK cell and IL7R+ NK cell was elevated while CD160+ NK cell was decreased in HCC samples (Fig. 2D). Furthermore, K-M survival curve suggested that FGFBP2+ NK cell exhibited an association with the prognosis of HCC patients (p =0.047), with higher survival probability in high ssGSEA score group in comparison to the low ssGSEA score group (Fig. 2E). Whereas, CD160+ NK cell (p =0.082) and IL7R+ NK cell (p =0.1) was not closely related to the prognosis of HCC patients (Fig. S2).

Figure 2 Analysis of natural killer (NK) cell heterogeneity.

(A) UMAP plot of NK cell subpopulations after annotation; (B) Highly expressed genes in each NK cell type; (C) Marker genes in each NK cell type; (D) Proportion of each NK cell type in different samples; (E) Kaplan–Meier (K–M) survival curve of ssGSEA score of FGFBP2+ NK cell in TCGA-LIHC cohort.

Pseudo-time trajectory of FGFBP2+ NK cell from normal to HCC was constructed

For further exploring the role of FGFBP2+ NK cell in HCC progression, the pseudo-time trajectory of FGFBP2+ NK cell from normal to HCC was constructed by Monocle2, and the branch with more healthy samples was set as the starting point, and the branch with more HCC samples was set as the ending point (Figs. 3A and 3B). Moreover, DEGs analysis between HCC samples and healthy samples revealed two differential expression gene clusters, and the gene expression in Cluster1 was gradually up-regulated with the increase of pseudo-time (Fig. 3C). KEGG enrichment analysis demonstrated that the DEGs in Cluster1 were primarily involved in the endocytosis, protein processing in nuclear factor (NF)-kappa B signaling pathway, endoplasmic reticulum, apoptosis, antigen processing and presentation, NK cell mediated cytotoxicity (Fig. 3D). These pathways might play a crucial role in the process of FGFBP2+ NK cells transitioning from the normal state to be involved in HCC.

Figure 3 Construction of pseudo-time trajectory of FGFBP2+ NK cell.

(A–B) Differentiation trajectory of FGFBP2+ NK cell from normal to HCC; (C) Heatmap of DEGs between HCC samples and healthy samples; (D) KEGG enrichment pathways of DEGs in Cluster1.

FGFBP2+ NK cell exhibited extensive intercellular communication in HCC

Cellular communication analysis was performed to explore the potential interactions between different cell types in HCC. It was observed that the number and strength of ligand–receptor interactions was complicated (Fig. 4A). FGFBP2+ NK cell as signal sender showed extensive communication with Macrophage, B cell, IL7R+ NK cell, T cell, CD160+ NK cell, and CD8+ T cell (Fig. 4B). By extracting the critical ligand–receptor pairs, we found that FGFBP2+ NK cell communicated with other cell subpopulations via tumor necrosis factor (TNF)-TNFRSF1B and macrophage migration inhibitory factor (MIF)-(CD74+CD44) (Fig. 4C), while other cell subpopulations communicated with FGFBP2+ NK cell via TNFSF14-TNFRSF14 and MIF-(CD74+CXCR4) (Fig. 4D).

Figure 4 Cellular communication analysis in HCC.

(A) Cell–cell interactions network between different cell subpopulations; (B) Cell communication of FGFBP2+ NK cell as a signal sender; (C) Key ligand–receptor pairs of FGFBP2+ NK cell acting on other cell subpopulations; (D) Key ligand–receptor pairs of other cell subpopulations acting on FGFBP2+ NK cell.

RiskScore model was constructed and verified based on 8-genes prognostic signature

Firstly, the marker genes in FGFBP2+ NK cell were subjected to univariate Cox regression analysis (p < 0.05). LASSO and stepwise regression analysis was further performed to reduce the gene number, and the model was optimal when the lambda was 0.0286 (Figs. 5A and 5B). Then, eight prognostic signatures related to FGFBP2+ NK cell in HCC were identified, including six “risk” genes (UBE2F, AHSA1, PTP4A2, CDKN2D, FTL, RGS2) and two “protective” genes (KLF2, GZMH) (Fig. 5C). Next, we established a RiskScore model of “RiskScore = −0.352*GZMH−0.303*KLF2+0.153*FTL+0.312*PTP4A2+0.442*UBE2F+ 0.234*CDKN2D+0.141*RGS2+0.427*AHSA1”.

Figure 5 Establishment and verification of RiskScore model.

(A) Coefficients of each independent variable; (B) Confidence interval for each lambda; (C) Prognostic gene signatures in RiskScore model; (D) RiskScore, survival status, and signature expression level in TCGA-LIHC cohort; (E) ROC analysis and K-M survival curves in TCGA-LIHC cohort; (F) RiskScore, survival Status, and signature expression level in ICGC-LIRI-JP cohort; (G) ROC analysis and K-M survival curves in ICGC-LIRI-JP cohort.

Furthermore, according to the threshold of RiskScore = 0, all samples in TCGA-LIHC cohort were separated into high- and low-risk groups (Fig. 5D). The performance of RiskScore model was evaluated by the area under ROC curve (AUC). It was showed that 1-, 3-, 5-years AUC values of the RiskScore model were 0.78, 0.77, 0.76 (Fig. 5E), which manifested that the RiskScore model exhibited good prognostic prediction performance. K-M curve analysis suggested that the OS rate of high-risk group was lower (Fig. 5E), showing that HCC patients with high RiskScore may have a poor prognosis. The robustness of RiskScore model was verified in ICGC-LIRI-JP cohort, and the results of which were similar to the TCGA-LIHC cohort (Figs. 5F and 5G). These outcomes demonstrated the reliability of the RiskScore model in the prognostic predicting of HCC.

RiskScore model exhibited potential in predicting immunotherapy response for HCC

We first elucidated the association between RiskScore and immune cell infiltration. ESTIMATE algorithm results showed that compared to low-risk group, high-risk group had lower StromalScore, ImmuneScore, and ESTIMATEScore (Fig. 6A). Based on the TIMER database, the infiltration levels of neutrophil, B cell, macrophage, and dendritic cell (DC) in high-risk group were significantly higher, while CD8 T cell was notably lower in comparison to the low-risk group (Fig. 6B). MCP-counter algorithm suggested that the infiltration levels of CD8 T cell, cytotoxic lymphocytes, NK cell, and endothelial cell were markedly lower, yet monocytic lineage was higher in high-risk group (Fig. 6C). In addition, the immunotherapy response between different risk groups was predicted in TCGA-LIHC cohort. In comparison to low-risk group, high-risk group exhibited higher exclusion score and TIDE score (Fig. 6D), demonstrating that high-risk HCC patients may be less likely to benefit from taking immunotherapy. RiskScore was positively correlated with several immune checkpoint genes, including CD44, CD276, CD80, LGALS9, and CTLA4 (Fig. 6E), showing that HCC patients with higher RiskScore may be more possibly to experience immune escape. Furthermore, we examined the association between drug sensitivity and RiskScore, and screened 16 drugs (such as Doramapimod, Nutlin.3a, Selumetinib, Sepantronium bromide, Tozasertib) that showed significant correlation with RiskScore (p < 0.05 and —cor— > 0.3) (Fig. 6F).

Figure 6 Correlation analysis between immune cell infiltration, immunotherapy response and RiskScore.

(A) StromalScore, ImmuneScore, and ESTIMATEScore between different risk groups; (B) Infiltration levels of six immune cells assessed by TIMER website; (C) Infiltration levels of 10 immune cells calculated by MCP-counter algorithm; (D) Exclusion, Dysfunction, and TIDE scores in different risk groups; (E) Relationship between RiskScore and immune checkpoint genes; (F) Correlation between RiskScore and drugs IC50; **** indicates p < 0.0001; *** indicates p < 0.001; ** indicates p < 0.01; * indicates p < 0.05; ns indicates not significant.

In vitro HCC cell-based model to validate key genes

The qRT-PCR analysis demonstrated that in comparison to the human hepatic astrocyte cells LX-2, the relative mRNA expression levels of GZMH was remarkably lower, yet FTL, PTP4A2, UBE2F, CDKN2D, RGS2, and AHSA1 were notably higher in HCC cells Huh7 (Fig. 7A). Since the pro-carcinogenic role of FTL in HCC has been reported in related studies, and the functional mechanism of PTP4A2 in HCC lacks systematic studies, we chose these two genes for in vitro functional experiments to further validate their roles in HCC cell migration, invasion and proliferation. Thereafter, we verified the knockdown efficiency of these two genes in Huh-7 cells (Fig. S3). We observed a significant decrease in proliferation of HCC cell lines after silencing FTL and PTP4A2 (Fig. 7B). In addition, wound healing assay displayed that FTL and PTP4A2 silencing could decrease the wound closure rate of Huh7 cells (Fig. 7C). Transwell assay suggested that the number of invading Huh7 cells was reduced by the silencing of FTL and PTP4A2 (Fig. 7D). Consequently, these results indicated the crucial involvement of FTL and PTP4A2 in HCC progression.

Figure 7 Effects of FTL and PTP4A2 silencing on the migration and invasion abilities of HCC cells.

(A) Relative mRNA expression of eight prognostic gene signatures detected by qRT-PCR; (B) Effect of FTL and PTP4A2 silencing on the proliferation ability of HCC cells was evaluated by EdU assay; (C) Impact of FTL and PTP4A2 silencing on the migration capability of HCC cells assessed via Wound healing assay; (D) Effect of FTL and PTP4A2 silencing on the invasion ability of HCC cells evaluated via Transwell assay; **** denotes p < 0.0001; *** denotes p < 0.001; ** denotes p < 0.01; * denotes p < 0.05.

Discussion

Numerous studies have indicated that NK cell function crucially in the surveillance and control of HCC, and NK cell dysfunction or exhaustion is implicated in the pathogenesis of advanced HCC (Eresen et al., 2024; Lee et al., 2021b). The prognostic significance of NK cell in TME of HCC has been widely concerned, and NK cell is highly considered as a promising target for tumor immunotherapy (Yang et al., 2023; Li et al., 2023). In this study, we innovatively combined scRNA-seq and bulk RNA-seq analyses to systematically resolve the heterogeneity of FGFBP2+ NK cells in HCC and its association with prognosis, and constructed a RiskScore model based on FGFBP2+ NK cell-related genes, which provides a new molecular basis for prognostic assessment of HCC patients and immunotherapy response prediction. Compared with previous studies, this study not only revealed the role of FGFBP2+ NK cells in the HCC microenvironment, but also explored the function of their related genes in HCC progression, revealing new potential targets for NK cell-targeted immunotherapy.

FGFBP2 served as the signature gene responsible for the cytotoxic killing function within NK cells (Li et al., 2010). Some researchers have identified FGFBP2 as a key gene involved in NK cell-mediated immune response in ankylosing spondylitis (Chen et al., 2025). Cui et al. (2024) identified two cell subpopulations in NK cells of HCC, including FGFBP2+ NK cells and B3GNT7+ NK cells. They suggested that the impairment of these functional anticancer cells is the potential cause of HCC. These studies suggest that the reduction of FGFBP2+ NK cells may impair the cytotoxic activity of NK cells, which in turn affects the immune response and prognosis of HCC patients, suggesting that it is expected to be a potential target for HCC immunotherapy. In addition, the eight prognostic signatures related to FGFBP2+ NK cell in HCC comprised six “risk” genes (UBE2F, AHSA1, PTP4A2, CDKN2D, FTL, and RGS2) and two “protective” genes (KLF2, and GZMH). In vitro assays showed that compared with the human hepatic astrocyte cells LX-2, the relative mRNA expressions of FTL, PTP4A2, UBE2F, CDKN2D, RGS2, and AHSA1 were notably higher in HCC cells Huh7. UBE2F, a ubiquitin-conjugating enzyme, is usually over-expressed in many malignant tumors such as lung cancer, resulting in a low OS rate (Zhou et al., 2020a). Targeting UBE2F might serve as an effective sensitizing strategy of chemo-/radiotherapy through triggering tumor cell apoptosis (Zhou et al., 2023). AHSA1, principally implicated in the activity of ATPase activators, is observed the upregulation of AHSA1 in HCC that is linked to clinical stage and worse outcomes of patients (Gao et al., 2023). Cell-based assays have manifested that knockdown of AHSA1 can inhibit the proliferation, migration, and invasion capabilities of HCC cells (Li & Liu, 2022). PTP4A2 belongs to a member of regenerating liver phosphatase family and can mediate cellular bioenergetics (Hardy et al., 2023). PTP4A2 is essential for vascular morphogenesis and angiogenesis (Poulet et al., 2020), which has been deemed as a carcinogenic factor in majority of human cancers, such as nasopharyngeal carcinoma (Gao et al., 2017) and glioblastoma (Chouleur et al., 2024). CDKN2D, a cyclin-dependent kinase inhibitor, exerts crucial roles in regulating tumor growth (Zhu et al., 2021). The abnormal expression of CDKN2D contributes to the uncontrolled proliferation of malignant cells including HCC (Lee et al., 2021a). FTL, also known as ferritin light chain, has been recognized to be one of the regulators in early iron metabolism (Kuang et al., 2023). Recent study of Ke et al. (2022) demonstrated that FTL was closely relevant to HCC progression, and patients with low FTL expression had a remarkable survival advantage. Our current study indicated that FTL silencing could markedly restrain the migration and invasion capabilities of HCC cells. RGS2, belonging to a GTPase-activating protein, functions as a key regulator of G-protein signaling (Cacan, 2017). Abnormal expression of RGS2 in different tumor types is significantly correlated with poor prognosis (Ihlow et al., 2022). There are few reports of RGS2 in HCC, deserving further exploration. KLF2, a transcription factor of Krüppel-like factor family, can impede the tumor cell movement mediated by TGF-β signaling in HCC (Li et al., 2020). Lin et al. (2019) observed that the expression of KLF2 was markedly decreased in HCC tissues than in adjacent tissues. GZMH, i.e., Granzyme H, exerts crucial roles in tumor killing mediated by T cell and NK cell, acting as a predictor for tumor immunotherapy (Li et al., 2024c). High GZMH expression is indicative of a better prognosis in HCC (Jin et al., 2024). Overall, these findings supported that the RiskScore model, constructed with eight prognostic signatures in this study, was reliable in the prognostic evaluation for patients with HCC. Meanwhile, these eight prognostic signatures may be regarded as promising therapeutic targets for HCC.

Furthermore, we found that the ImmuneScore in high-risk group was remarkably lower than that in low-risk group. The infiltration levels of B cell, neutrophil, macrophage, and monocytic lineage were observably higher while CD8 T cell, cytotoxic lymphocytes, NK cell, and endothelial cell were markedly lower in high-risk group than low-risk group. CD8 T cell plays an anti-tumor effect through releasing interferon-γ and TNF cytokines (Huo, Wu & Zang, 2021). Cytotoxic T lymphocytes exert crucial roles in anti-tumor immunity by recognizing and eliminating cancer cells (Akazawa et al., 2019). NK cell has strong anti-tumor activity via producing cytotoxic and cytokine (Zecca et al., 2020). This suggested that the infiltration characteristics of immune cells form an immunosuppressive microenvironment that may be closely correlated with a worse prognosis of HCC patients (Sun et al., 2023). The immunotherapy response in different risk groups was further predicted by TIDE algorithm, and high-risk group showed a higher TIDE score than low-risk HCC patients, which indicated that HCC patients in high-risk group might benefit limitedly from ICIs treatment (Liu et al., 2022). Moreover, the RiskScore exhibited positive correlation with several immune checkpoint genes, including CD44, CD276, CD80, LGALS9, and CTLA4, demonstrating that HCC patients with higher RiskScore may be more possibly to experience immune evasion (Xu et al., 2020a). Hence, these immune checkpoint genes could be the targets for HCC immunotherapy. In addition, the correlation between RiskScore and drug sensitivity was analyzed, and 16 drugs (such as Doramapimod, Nutlin.3a, Selumetinib, Sepantronium bromide, and Tozasertib) were screened, which supplied some reference for drug selection of HCC patients.

Nevertheless, there are also some shortcomings of the current study. The RiskScore model was constructed based on eight gene signatures, which was identified entirely from the data of public databases. It is necessary to further validate with prospective clinical data. Moreover, the specific mechanism of the eight prognostic gene signatures in HCC has not yet been investigated. In the future, we will consider performing a large number of in vivo and in vitro experiments to verify our outcomes.

Conclusion

In summary, based on bulk and scRNA-seq analysis, this study revealed eight prognostic gene signatures related to FGFBP2+ NK cell in HCC, which were utilized to create a RiskScore model. This model exhibited strong performance in assessing the prognostic outcomes and immunotherapy of HCC patients. 16 drugs were screened to be correlated with RiskScore. Additionally, FTL and PTP4A2 expression was upregulated in HCC cells, and their silencing significantly inhibited cell proliferation, migration, and invasive capacity. This study could provide promising therapeutic targets for HCC patients and also supply some reference for drug development.

Supplemental Information

Supplemental Information 1 UMAP plot of cell clusters in HCC before annotation

Supplemental Information 2 Kaplan–Meier (K–M) survival curves of ssGSEA score of CD160+ NK cell and IL7R+ NK cell in TCGA-LIHC cohort

Supplemental Information 3 Validate the knockout efficiency of FTL and PTP4A2 in Huh-7 cells based on qRT-PCR

Supplemental Information 4 Primer sequences used in qRT PCR

Supplemental Information 5 MIQE checklist

Abbreviations

API Application Programming Interface

AUC area under ROC curve

BNCC BeNa Culture Collection

cDNA complementary DNA

DC dendritic cell

DEGs differentially expressed genes

FBS fetal bovine serum

GDC Genomic Data Commons

GEO Gene Expression Omnibus

HCC hepatocellular carcinoma

IC50 half maximal inhibitory concentration

ICIs immune checkpoint inhibitors

KEGG Kyoto Encyclopedia of Genes and Genomes

K-M Kaplan–Meier

LASSO least absolute shrinkage and selection operator

MCP-counter microenvironment cell populations-counter

MIF Macrophage migration inhibitory factor

NF nuclear factor

NK Natural killer

OS overall survival

PCA principal component analysis

qRT-PCR quantitative real-time polymerase chain reaction

ROC receiver operating characteristic

scRNA-seq single-cell RNA sequencing

si small interfering

ssGSEA single-sample gene set enrichment analysis

TCGA The Cancer Genome Atlas

TIDE Tumor Immunity Dysfunction and Exclusion

TIMER Tumor Immune Estimation Resource

TME tumor microenvironment

TNF tumor necrosis factor

UMAP uniform manifold approximation and projection

Additional Information and Declarations

Competing Interests

Author Contributions

Data Availability

The authors declare there are no competing interests.

Yinbing Wu conceived and designed the experiments, performed the experiments, analyzed the data, prepared figures and/or tables, authored or reviewed drafts of the article, and approved the final draft.

Huanjun Peng conceived and designed the experiments, prepared figures and/or tables, and approved the final draft.

Guangkang Chen performed the experiments, analyzed the data, authored or reviewed drafts of the article, and approved the final draft.

Yinuo Tu performed the experiments, prepared figures and/or tables, and approved the final draft.

Xinpei Yu conceived and designed the experiments, analyzed the data, authored or reviewed drafts of the article, and approved the final draft.

The following information was supplied regarding data availability:

The datasets generated and/or analyzed during the current study are available at GEO: GSE162616.

The raw data is available in Github and Zenodo:

- https://github.com/yinuotu356/All-raw-data.git

- yinuotu356. (2025). yinuotu356/All-raw-data: Updated raw data (v.1.1.1). Zenodo. https://doi.org/10.5281/zenodo.14986495.

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
