# Peer review of "Bulk and single-cell RNA sequencing identify prognostic signatures related to FGFBP2+ NK cell in hepatocellular carcinoma"

_PeerJ, doi:10.7717/peerj.19337_

## Round 0.1 · original submission · Major Revisions

Based on the reviewers' evaluations, your manuscript requires major revisions before it can be considered for publication. Please carefully address all the comments and concerns raised by both reviewers. A detailed point-by-point response to the reviewers' comments should be included with your revised manuscript.

Reviewer 1 ·

Basic reporting

1. It should be noted that the cells were tested for mycoplasma contamination and found to be free of contamination, and that STR authentication was performed.
2. The Statistical analysis section lacks information on how to conduct statistical analysis for data that do not conform to a normal distribution, such as all the analyses in Fig. 7.
3. The data compared in Fig. 1c-d should be marked with significance symbols to indicate significant differences in the results.
4. In Fig. 2d, the main text describes the changes as increases or decreases in proportion. I suggest changing the y-axis to a percentage format, which would be more intuitive.
5. The results in Fig. 2e are poor. After six years, the survival rate of the high ssGSEA score group was worse than that of the low ssGSEA score group, and the p-value was very close to 0.05. The authors need to discuss the limitations of this study in the Discussion section.
6. "Yet FTL, PTP4A2, UBE2F, CDKN2D, RGS2, and AHSA1 were notably higher in HCC cells Huh7." First, add a comma before "yet" to improve readability. Here, we can see that FTL, PTP4A2, UBE2F, CDKN2D, RGS2, and AHSA1 were all significantly elevated in Huh7. So why was only FTL selected for the subsequent experiments?
7. After knocking down FTL, the efficiency of FTL knockdown should be proven with qPCR or Western blot results. Please supplement the experiments.
8. The results of the migration experiment need to be provided in high-resolution images because the cell outlines are not clearly visible.
9. The role of FTL in cell migration and invasion has been extensively studied. I believe that conducting only migration and invasion experiments is not very meaningful and lacks innovation.
10. It is not sufficient to only knock down FTL to test migration and invasion capabilities. I suggest that the authors select less-studied genes from the highly expressed genes in HCC cells and include proliferation and in vivo experiments. Such results would be more meaningful.
11. The Conclusion section should include all experimental results, including those from cell experiments.
12. The novelty of this paper is insufficient. I suggest that the authors highlight the strengths of this study in the Discussion section.

Experimental design

no comment

Validity of the findings

no comment

Reviewer 2 ·

Basic reporting

This study uncovers 8 prognostic signatures related to FGFBP2+ NK cell in hepatocellular carcinoma (HCC) through bulk and single-cell RNA sequencing, together with in vitro cell-based experiments, providing several potential therapeutic targets and drugs for HCC. The current study combines wet and dry methods with comprehensive experimental design. I recommend this article for publication, but the following deficiencies need to be improved before that.
1. In the “Establishment and verification of RiskScore model” section, the font size of “WERE CLASSIFIED” (Line146) is inconsistent with the full text. Please correct it to ensure consistency.
2. What are the specific parameters in the qRT-PCR amplification assay? It is necessary to clarify in the Methods part so as to facilitate the reader's reference and enhance the credibility of the article.
3. What are the characteristics and biological functions of FGFBP2+ NK cell? Are there any reports about FGFBP2+ NK cell in cancers especially HCC and its influence on the prognosis of patients?
4. In the introductory section, the full name needs to be defined when the abbreviation first appears, such as “HCC” (Line49). Please check the full text carefully and modify them.
5. The cellular communication analysis reveals several ligand-receptor pairs between FGFBP2+ NK cell and other cell subpopulations; what is the implications of this discovery? Detailed explanations are expected to be elaborated by author.
6. This study observes that the number of NK cell in HCC samples is markedly decreased, whether this is associated with the poor prognosis of patients?
7. PCR displays that the expressions of GZMH and KLF2 were downregulated while FTL, PTP4A2, UBE2F, CDKN2D, RGS2, and AHSA1 were up-regulated in HCC cells. Could the author elaborate on the appropriate reasons for choosing FTL silencing to explore the effects on the migration and invasion of HCC cell?
8. How many technical and biological replications were performed for in vitro assays of this study? Additionally, what statistical methods are used for in vitro (Figure 7)? They are required to be supplemented into the Methods section.
9. In Figure 1C and D, the position of annotations on the horizontal axis are suggested to realign to correspond to the columns as much as possible.

Experimental design

no comment

Validity of the findings

no comment

---

## Round 0.2 · accepted · Accept

Both reviewers have carefully evaluated your revised manuscript and find that you have adequately addressed their previous concerns. The revisions have significantly improved the quality of the paper. Based on the reviewers' recommendations, I am pleased to inform you that your manuscript has been accepted for publication.

Reviewer 1 ·

Basic reporting

no comment

Experimental design

no comment

Validity of the findings

no comment

Additional comments

In the new version, the author has added many experiments and made detailed supplementary replies to my questions one by one. The manuscript has been fully modified, and I think the current version meets the publication requirements of peerj.

Reviewer 2 ·

Basic reporting

This study uncovers 8 prognostic signatures related to FGFBP2+ NK cell in hepatocellular carcinoma (HCC) through bulk and single-cell RNA sequencing, together with in vitro cell-based experiments, providing several potential therapeutic targets and drugs for HCC. The current study combines wet and dry methods with comprehensive experimental design. I recommend this article for publication

Experimental design

no comment

Validity of the findings

no comment